# Environmental Risk Factors Contributing to the Spread of Antibiotic Resistance in West Africa

**DOI:** 10.3390/microorganisms13040951

**Published:** 2025-04-21

**Authors:** Adenike Adenaya, Adedapo Adedayo Adeniran, Chidera Linus Ugwuoke, Kaosara Saliu, Mariam Adewumi Raji, Amartya Rakshit, Mariana Ribas-Ribas, Martin Könneke

**Affiliations:** 1Institute for Chemistry and Biology of the Marine Environment (ICBM), University of Oldenburg, Carl von Ossietzky Str. 9-11, 26129 Oldenburg, Germany; amartya.rakshit@uni-oldenburg.de (A.R.); martin.koenneke@uni-oldenburg.de (M.K.); 2Department of Pharmacognosy and Natural Medicine, Faculty of Pharmacy, University of Calabar, Calabar 540211, Nigeria; adedapo.adeniran@unical.edu.ng; 3Department of Pharmaceutical Microbiology, Faculty of Pharmacy, University of Ibadan, Ibadan 240281, Nigeria; cugwuoke0078@stu.ui.edu.ng (C.L.U.); ksaliu0434@stu.ui.edu.ng (K.S.); mraji0375@stu.ui.edu.ng (M.A.R.); 4Center for Marine Sensors (ZfMarS), Institute for Chemistry and Biology of the Marine Environment (ICBM), Carl von Ossietzky University of Oldenburg, 26380 Wilhelmshaven, Germany; mariana.ribas.ribas@uni-oldenburg.de

**Keywords:** antibiotic resistance, global health problem, environmental pollution, Sub-Saharan Africa, West Africa, sanitation and hygiene, wastewater effluents

## Abstract

Antibiotic resistance is a well-documented global health challenge that disproportionately impacts low- and middle-income countries. In 2019, the number of deaths attributed to and associated with antibiotic resistance in Western Sub-Saharan Africa was approximately 27 and 115 per 100,000, respectively, higher than in other regions worldwide. Extensive research has consistently confirmed the persistent presence and spread of antibiotic resistance in hospitals, among livestock, within food supply chains, and across various environmental contexts. This review documents the environmental risk factors contributing to the spread of antibiotic resistance in West Africa. We collected studies from multiple West African countries using the Web of Science and PubMed databases. We screened them for factors associated with antibiotic-resistant bacteria and resistance genes between 2018 and 2024. Our findings indicate that antibiotic resistance remains a significant concern in West Africa, with environmental pollution and waste management identified as major factors in the proliferation of antibiotic-resistant bacteria and resistance genes between 2018 and 2024. Additional contributing factors include poor hygiene, the use of antibiotics in agriculture, aquaculture, and animal farming, and the transmission of antibiotic resistance within hospital settings. Unfortunately, the lack of comprehensive genetic characterization of antibiotic-resistant bacteria and resistance genes hinders a thorough understanding of this critical issue in the region. Since antibiotic resistance transcends national borders and can spread within and between countries, it is essential to understand the environmental risk factors driving its dissemination in West African countries. Such understanding will be instrumental in developing and recommending effective strategies nationally and internationally to combat antibiotic resistance.

## 1. Introduction

Antibiotic resistance represents a significant threat to human health, with the emergence and spread of multidrug-resistant (MDR) bacteria being one of our most pressing global health challenges [1]. Nine decades ago, antibiotics were developed to treat infectious diseases and prevent human deaths [2]. However, the global usage of antibiotics has increased, leading to antibiotic resistance and diminishing treatment options for infectious diseases and the prevention of deaths [3]. Antibiotic resistance, sometimes known as the “silent pandemic”, is a serious health concern, with about 1.2 million global deaths attributed to it in 2019 [4]. If prompt and adequate interventions are not in place, it might result in 10 million global deaths and also increase people′s disability-adjusted life years in the next two and a half decades [4,5]. Antibiotic resistance directly results from antibiotic consumption [6]. In 2017, the global antibiotic consumption reached 93 Mt and is expected to exceed 236 Mt by 2030 [7]. Indeed, human antibiotic consumption increased by 65% between 2000 and 2015, and a 200% increment is expected in the next five years [8]. Thus, despite antibiotics′ success in preventing and controlling infectious diseases, antibiotic resistance may pave the way for the next pandemic [9].

Antibiotic resistance presents a significant global challenge that affects all nations, irrespective of their level of development. However, the most severe impact and mortality rates related to this issue are predominantly felt in low-income countries, particularly in Sub-Saharan Africa. Murray et al. [4] indicated that, despite antibiotic resistance being a worldwide concern, Sub-Saharan Africa recorded the highest number of deaths attributed to antibiotic resistance in 2019 compared to other regions. From the data provided by the authors, it is estimated that Sub-Saharan Africa accounted for approximately 33% of the global burden of antibiotic resistance (Figure 1). While antibiotic resistance is also a pressing issue in various parts of Asia, the authors found that Western Sub-Saharan Africa experienced a notably higher burden, with around 18% of the global antibiotic resistance burden being carried by Western Sub-Saharan African countries. This implies that more than half of the antibiotic resistance burden within Sub-Saharan Africa is borne by this region alone. Approximately 115 deaths per 100,000 were estimated to be associated with bacterial resistance to multiple antibiotics in Western Sub-Saharan Africa in 2019 [4].

It is well-established that antibiotic resistance seriously threatens public health in West African countries. However, the environmental risk factors contributing to this issue have not been extensively studied, which hinders efforts to address the problem. Environmental risk factors encompass the factors facilitating the spread of contaminants in different environmental sources, including water, food, and air [10]. These factors are crucial in public health and safety [10]. They include pollution from untreated/poorly treated wastewater from different sources and other activities that contribute to the spread of antibiotic resistance in many low-income countries [11], including West Africa. Other environmental risk factors include the overall sanitary conditions of the country, the overuse and misuse of antibiotics, poor biosecurity measures, insufficient sanitation, the low economic status of the population, a lack of comprehensive management systems, inadequate public education, and poor overall education levels. In both clinical and natural environments, the overuse or misuse of antibiotics generates selective pressure, leading to antibiotic resistance. It is a natural phenomenon for bacteria to evolve resistance through two primary mechanisms: genetic mutations and the acquisition of resistance genes via genetic exchange in different environments [12].

Thus, a significant relationship between environmental factors and the spread of antibiotic resistance could exist. The interplay of these factors could foster the continual emergence of antibiotic resistance, which poses a persistent threat to human health. Antibiotic consumption can enhance horizontal gene transfer in pathogens, thus facilitating their spread and resistance capabilities between different environmental contexts [13]. However, despite the significant threat posed by antibiotic resistance, a notably disproportionate number of studies have been conducted on this topic in Africa, particularly in West Africa. In addition, many countries could not provide specific figures for deaths directly attributed to or associated with antibiotic resistance. To address this gap, Murray et al. [4] created predictive statistical models to estimate the impact of antibiotic resistance across all regions, including those with insufficient data.

A meta-analysis examining the proliferation of antibiotic resistance in food animals throughout Africa indicates that, although antibiotic resistance presents a considerable risk to food safety and security within the continent, a notably high number of studies addressing this problem are conducted in Nigeria [14]. Meanwhile, many low-income countries, including Nigeria, Ghana, Tanzania, Cameroon, Zambia, and Egypt, engage in extensive farming practices that utilize different antibiotics, including tetracycline, aminoglycosides, and penicillin groups, resulting in the spread of high numbers of MDR bacterial isolates [15]. A review by Ogunlaja et al. [16] highlighting the critical nature and the spread of antibiotic resistance in African aquatic environments also notes a disproportionate distribution of information from various regions across the continent. From a West African perspective, most studies have been conducted in Nigeria and Ghana, revealing a significant presence of antibiotic-resistance genes infiltrating aquatic environments, including tap and surface waters from different wastewater effluents in these countries [16]. Since antibiotic resistance transcends borders and can spread within and between countries, it is crucial to understand the environmental risk factors that influence its dissemination in Western Sub-Saharan African countries experiencing high mortality due to antibiotic resistance.

This review, therefore, explores the environmental risk factors contributing to the ongoing development and transmission of antibiotic resistance among humans, animals, and the environment in West Africa, including Cameroon, as it shares close borders with several West African countries, such as Guinea and Nigeria. Our research questions are as follows: (1) What are the primary sources of antibiotic-resistant bacteria and antibiotic-resistance genes in West Africa? (2) Which antibiotics are extensively studied? (3) Which bacterial strains and their associated antibiotic-resistance genes are prevalent in West Africa? (4) What environmental risk factors contribute to the spread of antibiotic resistance? Additionally, (5) what limitations and challenges do researchers in West Africa face in achieving an in-depth understanding of the spread of antibiotic resistance? The review, therefore, highlights the importance of various environmental elements, such as pollution, waste management challenges, inadequate sanitation facilities, and poor hygiene practices, in facilitating the continuous dissemination of antibiotic-resistant bacteria and antibiotic-resistance genes in West Africa. Furthermore, it addresses the factors leading to hospital-acquired resistance and the misuse of antibiotics in aquaculture and animal farming in this region.

## 2. Materials and Methods

Utilizing the Web of Science database and PubMed, we gathered a comprehensive selection of peer-reviewed articles focusing on bacterial antibiotic resistance in the West African region. Our search employed specific keywords such as “antibiotic resistance”, “antimicrobial resistance”, “drug resistance”, “resistance to antimicrobials”, “microbial resistance”, “resistance to antibiotics”, “multidrug resistance (MDR)”, “pathogen resistance”, “reduced drug susceptibility”, and “resistance to anti-infective agents”, coupled with terms like “environmental factors”, “ecological influences”, “environmental conditions”, “climatic factors”, “environmental determinants”, “habitat influences”, “ecological conditions”, “environmental variables”, and geographical identifiers including “West Africa”, “Western Africa”, “the West African region”, “ECOWAS region”, “Sub-Saharan West Africa”, “The Sahel region”, “The Niger Basin”, and “Guinea Region”. We additionally incorporated the names of individual West African countries in our search. To refine our results, we excluded literature reviews, systematic reviews, and gray literature and focused exclusively on open-access English research articles published between 2018 and 2024. Further screening was performed using the Covidence systematic review software (https://app.covidence.org) with defined inclusion and exclusion criteria, as shown in the PRISMA workflow presented in Figure 2.

## 3. Results and Discussion

### 3.1. Sources of Antibiotic Resistance in West Africa

Following our stringent inclusion and exclusion criteria (for instance, papers lacking molecular work and those published before 2018 were excluded), we incorporated 79 research papers in the current review (Figure 2). The number of studies retrieved from different countries varies according to our criteria. Notably, we did not acquire studies from several countries in the region. The majority of the included studies originated from the following countries: Benin Republic (6.3%, n = 5), Burkina Faso (3.8%, n = 3), Cameroon (5.1%, n = 4), Ghana (34.2%, n = 27), Niger (1.3%, n = 1), Nigeria (44.3%, n = 35), Senegal (1.3%, n = 1), The Gambia (2.5%, n = 2), and one study that involved both Benin Republic and Burkina Faso (1.3%). We could not obtain studies from Cabo Verde, the Ivory Coast, Guinea-Bissau, Liberia, Mali, Sierra Leone, and Togo. The substantial number of studies from Nigeria and Ghana corroborates the findings of Ogunlaja et al. [16], who pointed out in their review that Nigeria and Ghana contribute the most data on disseminating antibiotic-resistant bacteria and antibiotic-resistance genes in aquatic environments among Western Sub-Saharan countries. Founou et al. [14] noted similar trends regarding the spread of antibiotic resistance in food animals, with a lack of data from several countries, as also observed in this review.

Antibiotic resistance in the Western African region has been shown to affect humans, from pregnant women and their infants to individuals in the community [17,18,19]. It has been shown to be widespread in different contexts, including hospitals, communities, animal farms, food sources, and various environments (Figure 3). The environment, however, seems to be the primary source of antibiotic resistance, as shown by several studies in this review. Hospital settings, such as neonatology, pediatrics, maternity, operating rooms, and sterilization units, demonstrate a significant prevalence of *Staphylococcus aureus* strains harboring the *mecA* gene [20]. This can be attributed to the proximity of patients and the shared resources within the hospital environment, which facilitate the transmission and persistence of this antibiotic-resistant pathogen throughout the hospital wards [20]. Hospital items such as bed wheels, bedpans, trays, bed handles, mattresses, pillows, bedside carts, the tops and drawers of bedside cabinets, desk surfaces, waiting areas, chair handles, thermometers, stethoscopes, sphygmomanometers, computer keyboards, window levers, light switches, wall socket switches, tap handles, and washroom door handles have been found to harbor MDR bacteria, particularly *Acinetobacter baumannii* [21]. This bacterium carries the New Delhi metallo-beta-lactamase 1 (bla_NDM-1_) gene, and Acolatse et al. [21] attributed the spread of this resistance to insufficient infection prevention and control practices within hospital settings.

Inadequate food processing practices allow the colonization of different foods with antibiotic-resistant bacteria and resistance genes. Antibiotic-resistant bacteria can spread from meat to humans during animal food processing, contaminating processing equipment or storage areas [22]. As such, objects like knives, tables, and cooking utensils are not left out [23,24]. Animal feces carry resistant germs and contaminate the environment and food, such as dairy products [25]. Fruits and vegetables have been contaminated during processing and when they come into contact with wastewater and fertilizer containing animal and human waste during planting [26,27]. 

Furthermore, there is soil contamination from human and animal fecal matter [25] and poultry feces used as fish feed, as bird excreta can contain pathogenic antibiotic-resistant bacteria such as *Escherichia coli*, *Pseudomonas* spp., *Enterobacter* spp., *Acinetobacter* spp., and many more [28]. Effluents from different sources, different aquatic environments, and other environmental compartments, such as household dust and dumpsites, harbor many antibiotic-resistant bacteria and antibiotic-resistance genes [29,30,31,32,33,34,35,36]. Antibiotic-resistant bacteria from these sources are also common in other African countries, such as South Africa [37]. In high-income countries, hospitals and municipal wastewater are the primary sources of antibiotic-resistant bacteria [38,39]. Nonetheless, the lack of regulation regarding antibiotic use in Sub-Saharan African countries has exacerbated the development and spread of antibiotic resistance [40].

### 3.2. Antibiotics Under Study in West Africa

Commonly studied antibiotics in hospitals, food, and environments were those belonging to the beta-lactams, quinolones, cephalosporins, aminoglycosides, tetracyclines, carbapenems, sulfadimidine, phenols, and macrolides classes. Ampicillin and amoxicillin-clavulanate are the predominant beta-lactam antibiotics under investigation. Notably, cefotaxime, ceftazidime, and ceftriaxone have been extensively researched within the cephalosporin class. Their inappropriate use across various environmental contexts is closely linked to rising antibiotic resistance in West Africa. In Ghana, the practice of self-medication with amoxicillin is prevalent among the sick populace [41]. Data from hospital surveys in rural Ghana reveal that ceftriaxone and cefuroxime were among the top five antibiotics prescribed [42]. In Gambia, ampicillin and ceftriaxone are among the antibiotics used for empirical first-line treatment of hospitalized infants (age < 72 h) [17]. Nigeria and Ghana, in particular, have witnessed the extensive use of beta-lactam antibiotics in agricultural practices [43,44,45], leading to the widespread beta-lactam resistance in animal farms.

Ciprofloxacin is the most extensively studied antibiotic among the quinolone class, likely due to its effectiveness against various bacterial strains, including *Escherichia coli* and *Vibrio* spp. [46,47,48]. This broad-spectrum efficacy has led to its widespread use in hospitals, aquaculture, and animal husbandry. According to Bah et al. [17], ciprofloxacin is the second line of treatment for hospital-acquired multidrug-resistant gram-negative bacteria (MDR-GNB). However, the resistance of extended-spectrum beta-lactamase (ESBL) producing *Enterobacteriaceae* toward ciprofloxacin could be high, particularly among young children [49]. Quarcoo et al. [27] noted significant resistance to ciprofloxacin among *Escherichia coli* isolates obtained from irrigation waters contaminated with untreated/poorly treated wastewater runoff, indicating that the environment plays a substantial role in ciprofloxacin resistance. Reports of high concentrations of ciprofloxacin found in the urine of healthy individuals in Ghana suggest chronic exposure, potentially coming from contaminated food sources such as fish and dairy products [19].

Tetracycline antibiotics are extensively studied and are commonly utilized in aquaculture, especially in freshwater fish farming, which is prevalent in developing countries like Nigeria [50,51]. Tetracycline is readily available over the counter and is often administered through feeds or baths. Tetracycline was identified as one of the most overused drugs in the Nigerian broiler production value chain [52]. Within the macrolide category, erythromycin and azithromycin emerge as the most frequently studied. Erythromycin’s application is particularly pronounced in live bird markets, where adherence to antibiotic withdrawal periods is often lacking [52]. Bird excreta frequently contain antibiotic-resistant bacteria and antibiotic-resistance genes, which can lead to soil contamination and subsequent washing into nearby surface waters. Consequently, it is unsurprising that antibiotic resistance is exceptionally high in the affected environments, as observed in several studies. Carbapenems are considered the last line of defense against multidrug-resistant bacteria. However, the spread of carbapenemase-producing bacteria, particularly *Escherichia coli* and *Klebsiella pneumoniae*, and the distribution of the resistance genes in various environmental contexts, including humans [53], animals [54], food [24], hospital setting [55], hospital effluents [56], coastal environments [57], freshwater [58,59] and dumpsites [12], are of significant concern.

### 3.3. Antibiotic-Resistant Bacterial Strains, Along with Their Associated Resistance Genes

The key pathogens in our review include *Escherichia coli*, *Klebsiella* spp., *Pseudomonas* spp., *Enterobacter* spp., *Staphylococcus* spp., *Acinetobacter* spp., *Citrobacter* spp., *Bacillus* spp., *Aeromonas* spp., *Proteus* spp., *Salmonella* spp., *Serratia* spp., *Shigella* spp., *Vibrio* spp., and *Enterococcus* spp. (Appendix A). In many West African countries, the spread of *Escherichia coli* and *Klebsiella* spp. is the order of the day, as they have been found in different settings, including hospitals and community settings, on vegetables, chickens, and pigs, and municipal solid waste dumpsites, groundwater, and surface water [12,26,53,58,59,60,61]. Murray et al. [4] showed that more than 15% of deaths were attributed to these pathogens in 2019. *Escherichia coli* was found to be associated with various infectious diseases, such as urinary tract infections, neonatal sepsis, and diarrheal illnesses. For instance, Adegoke et al. [62] studied the prevalence of urinary tract diseases among hospitalized patients. They found higher numbers of *Escherichia coli* in the patient′s urine, indicating the presence of such diseases, as healthy urine is considered sterile and should not contain bacteria [62]. In the southwestern part of Nigeria, meat and meat sellers′ hands have been found to harbor numerous MDR *Escherichia coli* strains that show 100% resistance to ceftazidime, cefotaxime, and amoxicillin-clavulanic acid [63]. Some strains were studied to produce ESBL.

About 50% and 20% of ESBL-producing *Escherichia coli* are widespread in Benin hospital effluents and fecal samples [18]. Many ESBL-producing *Escherichia coli* and *Klebsiella* spp. in northern Cameroon, showing a high co-resistance pattern to common antibiotics, have been detected in the urine of infants (0 to 3 months) diagnosed with urinary tract infections [49]. MDR and ESBL-producing *Escherichia coli* have been isolated from different clinical samples in the Benin Republic and Nigerian hospitals [18,64]. Approximately 77% of ESBL-producing *Klebsiella* spp. were retrieved from infants admitted to various hospitals in Accra, Ghana, with 7% developing bloodstream infections [65]. ESBL-producing *Escherichia coli, Klebsiella pneumoniae*, and other *Enterobacteriaceae* harbor several antibiotic-resistance genes, including *bla_TEM_, bla_SHV_*, and *bla_CTX-M_* [53,55,66,67,68,69], allowing them to express their physiological resistance to beta-lactams and other antibiotics. The *bla_TEM_* gene, particularly the *bla_TEM__-1B_* variant, produces beta-lactamases that hydrolyze penicillin and other beta-lactam antibiotics [70]. The bla_SHV_ gene is associated with extended-spectrum cephalosporin resistance and is mostly found in *Enterobacteriaceae*. Beta-lactamase is a group of enzymes produced by certain bacteria that confer resistance to a wide range of beta-lactam antibiotics, including extended-spectrum cephalosporins, penicillins, and aztreonam [71].

The prevalence of *bla_CTX__-__M_*, a critical family of beta-lactamases that confer resistance to third-generation cephalosporins, was documented in *Escherichia coli and Klebsiella pneumoniae* obtained from major hospitals in southeastern Nigeria [69]. *Escherichia coli, Klebsiella pneumoniae*, and various other members of the *Enterobacteriaceae* family also harbor carbapenem-resistant genes such as bla_TEM-1B_, bla_CTX-M-15_, and bla_OXA-1_. These resistance elements have been isolated from diverse reservoirs, including humans, animals, and environmental sources [12], highlighting the widespread nature of antibiotic resistance within these bacterial populations in West Africa. Other antibiotic-resistant bacteria and resistance genes, such as colistin-resistant strains and methicillin-resistant *Staphylococcus aureus* (MRSA), are widespread in this region, and several factors contribute to their prevalence.

Colistin-resistant bacteria were retrieved in 70% of 4907 rectal samples collected from infants and their mothers across various hospitals in Kano and Abuja, Nigeria [72]. Portal et al. [72] noted that the infants were less than one week old, and neither they nor their mothers had previously received colistin treatments. In addition, seven infants and 40 mothers were found to harbor colistin-resistant genes (*mcr*). Isolates of *Enterobacter* spp., *Shigella*, *Escherichia coli*, and *Klebsiella quasipneumonia* carrying these genes were obtained from some samples. The authors suggested that the mothers may have acquired these colistin-resistant bacteria from their surrounding environment. Mohamadou et al. [73] reported that approximately 46% (n = 202) of MRSA cases were found among young children in various hospitals across the northern regions of Cameroon. The presence of antibiotic-resistance genes, such as *mecA*, contributed to the prevalence of MRSA strains, particularly in clinical settings in Cameroon [73]. The authors noted that these bacterial strains exhibited high resistance to multiple antibiotics, including cotrimoxazole, gentamicin, ofloxacin, penicillin, and tetracycline. They attributed the significant number of bacterial isolates to infrequent hospitalizations, indicating that the infections were not necessarily acquired in hospital settings, even though MRSA typically originates from such environments. Furthermore, the authors emphasized that environmental factors such as poor sanitation and personal hygiene contributed to the spread of these infections, as MRSA was reported to be easily transmitted through hand contact. Thus, the spread of antibiotic resistance can be attributed to several environmental risk factors.

### 3.4. Environmental Risk Factors Contributing to the Spread of Antibiotic Resistance in West Africa

While the specific environmental risk factors contributing to the spread of antibiotics were not directly investigated, they have been linked to the ongoing acquisition and dissemination of antibiotic resistance in West Africa, as noted in most papers reviewed in this study. These factors can be categorized into four primary areas: environmental pollution and waste management, poor hygiene, agriculture, aquaculture, and animal antibiotic use, and transmission within hospital settings. There is also a combination of the first two factors (Figure 4). Environmental pollution and waste management were identified as the most common environmental risk factors, reported in 39% of the studies. This issue is mainly associated with the discharge of untreated/poorly treated wastewater and open defecation. Such practices can lead to the contamination of surface water, groundwater, and food sources. Furthermore, coliform bacteria, such as *Escherichia coli*, indicate fecal contamination in these sources.

Fifteen percent of the studies highlighted poor hygiene as a significant factor contributing to the ongoing spread of antibiotic-resistant bacteria and resistance genes. This issue is mainly associated with the poor hygiene of most people, particularly food handlers who contaminate food products. 18% of the studies addressed antibiotic use in agriculture, aquaculture, and animal settings, and 17% indicated the transmission of resistance within hospitals. The former is associated with the indiscriminate use of antibiotics in fish and animal farming, while the latter is associated with hospital-acquired antibiotic-resistant infections. The remaining 11% combines environmental pollution, waste management, and poor hygiene. This confirms that the two environmental risk factors play key roles in acquiring, transferring, and disseminating antibiotic-resistant bacteria and resistance genes in West Africa. We discuss each of these factors in the following sections.

#### 3.4.1. Environmental Pollution and Waste Management

The discharge of untreated/poorly treated hospital effluents into the environment, along with the improper disposal of untreated domestic sewage into surface waters, represents significant environmental pollution and waste management challenges [29,30,33,56,59,68,74,75]. Many studies gathered samples from various environmental sources, including domestic, animal, hospital, and industrial wastewater; freshwater; groundwater; coastal sediments and wetlands; agricultural soil; irrigation water; air; household dust; and dumpsites (Figure 5). The inefficient treatment of hospital wastewater constitutes the most frequent environmental pollution. According to Oladipo et al. [74], hospital effluents are a significant reservoir for antibiotic-resistant bacteria and resistance genes. The authors found that hospital effluents discharged into the environment contain resistant *Staphylococcus* species, including MRSA, with a high prevalence of the *mecA* genes. Their study highlights the potential role of hospital wastewater in disseminating antibiotic-resistant bacteria and resistance genes into natural water bodies, posing significant risks to public and environmental health in Nigeria. These genes were often associated with mobile genetic elements, indicating horizontal gene transfer to other bacteria in the environments [76].

Hospital effluents in Ghana were identified as a significant reservoir for MDR *Escherichia coli* strains, which exhibited resistance to a range of antibiotics, including ampicillin, meropenem, azithromycin, and sulfamethoxazole–trimethoprim [76]. Indeed, beta-lactamase genes (e.g., *bla_TEM-1B_, bla_CTX-M-15_*) and carbapenemase genes (e.g., *bla_OXA-181_*) are prevalent in *Escherichia coli* obtained in the effluent. Hospital effluents in Ouagadougou, Burkina Faso, were found to contain carbapenemase-producing *Escherichia coli* and *Klebsiella pneumoniae*, which were resistant to multiple antibiotics, including last-resort carbapenems like ertapenem [30]. Kagambèga et al. [31] indicated that *Escherichia coli* and *Klebsiella pneumoniae* resistance rates to ciprofloxacin, ampicillin, and other beta-lactams were exceptionally high, probably due to various beta-lactamase genes, including *bla_NDM_, bla_VIM_, bla_IMP_, bla_KPC_*, and *bla_OXA-48_*, detected in them.

Discharging untreated domestic sewage into surface waters and open defecation, particularly in conjunction with insufficient municipal solid waste disposal, contributes immensely to these problems [77,78]. *Escherichia coli* is an indicator of fecal contamination caused by the improper disposal of untreated domestic sewage into surface waters and open defecation. Banu et al. [59] investigated the contamination levels in various rivers within two major cities in Ghana. The authors found many *Escherichia coli*, and most of their isolates could produce beta-lactamase genes. They observed that these rivers are near open markets, shopping centers, and slums, where open defecation practices are common along their banks. They advocated for improvements in basic sanitation infrastructure and household toilets to help reduce open defecation among the population. Over 60% of the people in a major city in Ghana reportedly dispose of their feces alongside household wastes, in gutters, or within drainage systems [79]. Open defecation is also a big challenge in Nigeria. According to Abubakar [80], about 46 million people in Nigeria practiced open defecation in 2015. In southwestern Nigeria, the prevalence of open defecation in 2018 was more than 20% [81], and the practice is not ending soon [82].

Open defecation and inappropriate disposal of feces create hotspots for the acquisition, transfer, and dissemination of antibiotic-resistance genes [11,76]. It is, therefore, unsurprising that antibiotic-resistant *Vibrio cholerae* type O1 has been isolated from household sources, including shallow wells, storage tanks, and tap water in Ghana [77]. Indeed, 93% and 73% of the strains were resistant to erythromycin and nalidixic acid, respectively. However, the strain was found to be susceptible to ciprofloxacin and oxytetracycline [77]. *Vibrio* spp. harboring various antibiotic-resistance genes are prevalent in many rivers across southwestern Nigeria [83,84]. The occurrence of cholera in several states in Nigeria is becoming increasingly common. Most *Vibrio* spp. obtained from stools and seafood in coastal areas were shown to be resistant to amoxicillin–clavulanic acid, amikacin, and colistin [85]. Fortunately, ciprofloxacin was still effective against *Vibrio cholerae* in this environment [85].

Sometimes, feces are collected in chamber pots, diapers, and plastic bags and disposed of in municipal solid waste landfills, making this environment a hotspot for antibiotic-resistant bacteria and antibiotic-resistance genes [79,86]. Municipal solid waste landfills often contain a variety of pollutants, including plastic waste such as water bottles and sachets [87,88]. These waste disposal sites also contain sanitary tissue papers, unused pharmaceuticals, personal healthcare products, and hospital and industrial wastes [61]. The liquid that forms when water percolates through solid waste landfills, known as leachate, has been shown to contain antibiotic compounds, various antibiotic-resistant pathogens, and antibiotic-resistance genes. For instance, leachates collected from four distinct solid waste landfills in Kumasi, Ghana, revealed high concentrations of penicillin (67 ± 5 μg mL^−1^), amoxicillin (10 ± 6 μg mL^−1^), and metronidazole (18 ± 7 μg mL^−1^) [89].

Adekanmbi et al. [61] retrieved 250 cefotaxime-resistant bacteria from dumpsite leachates obtained in southwestern Nigeria, and all of them could produce ESBL. According to the authors, more than 75% of the bacteria have different variations of beta-lactam-resistance genes, including the *bla_CTX-M-2_, bla_CTX-M-8_*, and *bla_CTX-M-25_*. Adekanmbi et al. [34] also found bacterial isolates resistant to third-generation cephalosporins, including cefotaxime (100%), cefpodoxime (97%), and ceftazidime (97%), from dumpsite leachates and the surrounding surface water, exacerbating antibiotic resistance spread through improper waste management. Other anthropogenic activities relating to environmental pollution and waste management include farming and industrial practices along watercourses, discharging animal and industrial wastes into surface water, and contamination of irrigation water sources with untreated/poorly treated wastewater [66,90,91]. These practices raise significant concerns regarding public health, environmental integrity, environmental sanitation, and hygiene.

#### 3.4.2. Poor Hygiene

This involves contamination of food, food items, and various surfaces due to unwashed hands, which facilitates the distribution of antibiotic-resistant genes in humans, animals, and the environment (Figure 5). Several studies provide a compelling case for addressing food hygiene and safety challenges in West African countries and advocate for preventive measures to curb the spread of antibiotic resistance and foodborne diseases through this route. For instance, Abas et al. [63] investigated the contamination of majorly consumed street food like *Tuo-Zaafi* (made from maize flour) from different food vendors in Ghana. They found that about 50% of the food samples collected were contaminated with *Escherichia coli*, *Shigella* spp., *Salmonella* spp., and *Staphylococcus aureus*, which are frequently associated with poor hygiene. The resistance of these isolates to commonly used antibiotics was, in most cases, more than 80%. According to the authors, the initial food contamination resulted from the vendors’ poor hygiene, which they also confirmed by visiting vendors’ kitchens. The authors said, “Most of these people did not wash their hands between different activities at the site, including handling money and using the washroom before handling the food” [63]. Similarly, *Escherichia coli* was prevalent on the hands of meat handlers such as retailers (53%) and butchers (87%) in different communities in Southern Nigeria [46]. The isolates contaminate knives, tables, and floors of retail shops, and about 50% of them harbor antibiotic-resistance genes, such as the beta-lactamase genes [46]. The lack of food safety regulations in many African countries contributes to the ongoing problem of antibiotic resistance.

According to Somda et al. [46], a lack of hygiene practices during the slaughter, preparation, grilling, and handling of ready-to-eat chicken consumed by many people in Burkina Faso significantly increases contamination risks by fecal coliforms in several outlets. The author retrieved diarrheagenic *Escherichia coli* strains with resistance to ampicillin (43%), amoxicillin/clavulanic acid (46%), and tetracycline (64%) from the chickens and concluded that implementing good hygienic practices at slaughterhouses and sales outlets could reduce bacterial contamination and the widespread antibiotic resistance. In Cameroon, the spread of *Staphylococcus* species, including MRSA and methicillin-resistance genes, was linked to poor hygiene and sociocultural practices among approximately 60% of respondents [73]. Another study from Cameroon indicated that pigs and people employed in slaughterhouses harbor various strains of *Klebsiella pneumoniae*. These strains were found to possess resistance genes targeting a range of antibiotic classes, including beta-lactams, aminoglycosides, fluoroquinolones, macrolides, and others [92]. *Vibrio parahaemolyticus* with high resistance to tetracycline, kanamycin, and chloramphenicol and resistance genes to these antibiotics were isolated from several ready-to-eat food outlets in Nigeria [93]. Packaged vegetables and salads from several retail outlets in Nigeria were found to harbor pathogenic strains of *Escherichia coli*, *Klebsiella pneumoniae*, and *Klebsiella variicola* with more than 90% resistance to ceftriaxone, cefotaxime, ceftazidime, aztreonam, ampicillin, and piperacillin [94].

In some cases, hygiene standards in food preparation were not complied with [24]. For instance, in the Benin Republic, Dougnon et al. [24] revealed that community kitchen samples exhibited the highest prevalence of bacterial contamination, reaching 36% among all sample types. Notably, 47% of the bacteria isolated from these environments were identified as producers of ESBL. Additionally, the *bla_SHV_* gene was detected in 33% of the isolates. The level of multidrug resistance observed was high, with many bacteria exhibiting resistance to multiple classes of antibiotics. Their findings suggest that non-compliance with hygiene standards in food preparation and handling creates conditions that facilitate the proliferation and transfer of antibiotic-resistant bacteria and genes, which could be detrimental in highly urbanized areas.

Inadequate sanitation in urban and peri-urban areas of West Africa often creates conditions that facilitate the transmission of antibiotic-resistant bacteria. Darboe et al. [95] linked high population to the proliferation of virulent strains, particularly the Panton-Valentine Leukocidin (PVL) genes in PVL-positive *Staphylococcus aureus*, especially in densely populated cities like Banjul in Gambia. Over 11 years, the authors investigated the prevalence and antibiotic resistance of *Staphylococcus aureus* isolated from communities and patients at an urban Gambian hospital. Their findings revealed that approximately 61% of the isolates carried PVL genes, with a higher prevalence of invasive infections (72%) than non-invasive cases (57%). Interestingly, methicillin resistance was relatively low at 2%, and resistance rates for other antibiotics were minimal, although resistance to penicillin was very high at 92%. Antibiotic-resistant *Staphylococcus* spp. could colonize healthy individuals with chronic antibiotic exposure, particularly in communities with high populations [19]. The spread of *Staphylococcus* spp. in these healthy individuals was attributed to the use of antibiotics in aquaculture and animal farms, such as dairy and fish products in Ghana [19].

#### 3.4.3. Agriculture, Aquaculture, and Animal Antibiotic Use

No studies indicate the direct use of antibiotics in agriculture; however, they showed that untreated/poorly treated wastewater often used for vegetable irrigation is the reservoir of antibiotic-resistant bacteria and resistance genes. Vegetables are critical sources of nutrition, with numerous campaigns highlighting their significance in combating various diseases [93]. These initiatives have increased the availability of ready-to-eat vegetables in several regions worldwide, including West African countries. However, contaminated irrigation water sources with human and animal wastes were studied to be used for the irrigation of vegetable farms in most West African countries [27,43,47,96].

Generally, in most Sub-Saharan Africa, a significant challenge persists regarding the availability of safe water, forcing the population to rely on contaminated water sources [97]. The lack of safe water facilitates using untreated/poorly treated wastewater for irrigation [98]. In Ghana, open drains and hosepipes of untreated/poorly treated wastewater were used for irrigation. The wastewater yielded the highest bacterial counts, highlighting contamination during irrigation and subsequent contamination of vegetables with *Escherichia coli* [27]. A study by Appau and Ofori [43] showed that irrigation water containing poultry waste used for vegetables at various sampling sites in Kumasi, Ghana, as well as the associated soil, harbors at least 3.9 × 10⁶ coliforms that exhibit significant levels of antibiotic resistance. Specifically, 94% of these coliforms were resistant to meropenem, 92% to ampicillin, 95% to cefuroxime, 94% to ceftriaxone, and 94% to cefotaxime. The authors suggest that the use of antibiotics in poultry and the subsequent application of poultry waste serve as significant sources of antibiotic-resistant bacteria and resistance genes.

Chigor et al. [47] investigated the microbiological quality of treated wastewater used for irrigation and its role in the spread of enterotoxigenic and MDR coliform *Escherichia coli* in southern Nigeria. In addition to treated wastewater samples, they analyzed irrigated soil and vegetables for bacterial identification and antibiotic susceptibility testing. The authors found that all treated wastewater, soil, and vegetables tested positive for coliform *Escherichia coli*, with 44% of the vegetable isolates containing the heat-labile toxin (LT) genes. This indicates that these strains are enterotoxigenic and capable of causing diarrhea in humans when the vegetables are consumed. Additionally, the isolates in the vegetables exhibited high resistance to several antibiotics, including penicillin, vancomycin, and clarithromycin.

Another factor contributing to the persistent threat of antibiotic resistance through this route is the lack of regulation of antibiotic use in fish and animal farms. While this issue spans most low- and middle-income countries, most studies regarding the indiscriminate use of antibiotics in fish and animal farms in this review originate from Nigeria, indicating that the problem is particularly severe in the region. This could be because Nigeria has experienced notable growth in the aquaculture sector over the past two decades. Currently, Nigeria is the largest aquaculture producer in Sub-Saharan Africa [99] and the second largest in the entire continent, with fish farming rapidly becoming one of the fastest-growing agricultural enterprises [100]. The consumption of aquaculturally produced fish, mainly African catfish (*Clarias gariepinus*), is pervasive, with nearly every household incorporating it into their diet. Additionally, these fish are readily available in various commercial establishments, including hotels, restaurants, and pubs. They are frequently featured as a menu item at private events, indicating their significance in everyday meals and special occasions. However, the lack of stringent regulations regarding antibiotic use in aquacultural production has raised concerns about potential health risks to fish consumers [51].

In Northern Central Nigeria, it has been reported that approximately 78% of farmers utilize antibiotics in aquaculture production [101]. Furthermore, 94% of these farmers use self-prescribing antibiotics, highlighting a significant concern regarding the lack of professional guidance in treating fish. Additionally, 95% of the farmers do not adhere to withdrawal periods following antibiotic use [101]. It might be assumed that a lack of education is the primary issue; however, research has indicated that fish farming is often regarded as an elitist occupation in Nigeria, appealing to individuals with higher levels of education compared to other forms of animal farming [102]. Adah et al. [103] highlight the significance of implementing stricter regulatory measures and enhancing aquaculture practices to combat the spread of MDR pathogens in Nigerian aquaculture production. They carried out the molecular characterization and antibiotic resistance of *Aeromonas* spp. isolated from farmed *Clarias gariepinus*. Their findings indicate significant resistance to commonly used antibiotics, including oxytetracycline, penicillin, and colistin sulfate, with a multiple antibiotic resistance index ranging from 0.2 to 0.8, suggesting widespread misuse of antibiotics in aquaculture. Antibiotics used for fish are not metabolized, releasing them into the surroundings [101].

Animal farms harbor bacteria resistant to carbapenems and colistin [12]. This trend highlights the extensive application of these antibiotics in animal farming practices, raising significant concerns regarding public health and the potential for zoonotic transmission of resistance genes [12]. Oloso et al. [52] provide critical insights into the inappropriate use of antibiotics within Nigeria’s broiler production value chain and its consequences for antibiotic resistance. Their findings indicate a widespread and indiscriminate application of antibiotics, with over 80% of instances occurring without veterinary oversight. According to the authors, the antibiotics commonly used in Nigeria′s broiler production are enrofloxacin, tetracycline, and erythromycin, which are administered prophylactically and metaphylactically. They also studied the antibiotic resistance patterns of *Salmonella* spp., originating from broiler production, and found high resistance rates of 95% and 100% for older-generation antibiotics such as penicillin and flumequine, respectively. Again, it may be presumed that insufficient educational opportunities constitute the primary concern regarding antibiotic usage among poultry farmers. However, approximately 95% of poultry farmers in Nigeria have a solid understanding of how to apply antibiotics, with 83% of this cohort exhibiting an awareness of the development of antibiotic resistance in this way [104]. Oloso et al. [52] highlight the exacerbating effect of misusing newer-generation drugs, like enrofloxacin, on the resistance observed in older formulations in animal farming. Their study emphasizes the harmful impact of inadequate biosecurity measures, unregulated antibiotic access, and substandard farming practices in the spread of antibiotic resistance in animals and humans.

#### 3.4.4. Transmission Within Hospital Settings

The World Health Organization [105] identifies antibiotic-resistant, hospital-acquired infections (nosocomial infections) as a leading cause of morbidity and mortality globally. They estimate that around 64% of antibiotic-resistant infections are associated with hospital settings, resulting in 136 million antibiotic-resistant infections acquired in hospitals worldwide. This issue is particularly critical in low- and middle-income countries, where limited resources and infrastructure impede effective infection prevention and control measures. However, high-income countries are not immune, as demonstrated by outbreaks of resistant pathogens in advanced hospital settings [105].

Indeed, numerous studies in this review revealed that hospitals serve as significant sources of antibiotic-resistant bacteria and resistance genes across many West African nations, suggesting that antibiotic resistance is frequently acquired within hospital environments. This is also true in high-income countries as well as in several Asian countries [4]. Antibiotic resistance nosocomial infections are primarily caused by opportunistic pathogens, including *Klebsiella pneumoniae*, *Escherichia coli*, *Staphylococcus aureus, Pseudomonas aeruginosa*, and *Acinetobacter baumannii*, that thrive in such settings [13,17,73]. These organisms often exhibit resistance to multiple antibiotics, complicating treatment efforts and resulting in prolonged hospital stays, increased healthcare costs, and elevated mortality rates [105].

Bah et al. [17] suggests that the hospital environment, rather than vertical transmission (transmission of antibiotic resistance from a mother to her baby) plays a significant role in neonatal colonization. They assessed the prevalence and dynamics of MDR-GNB among neonates with prolonged hospital stays in Gambia. They found that 41% of neonates carried at least one strain of MDR-GNB upon admission. After seven days, all surviving neonates had developed MDR-GNB, with 85% acquiring new strains during their hospitalization. The authors indicated that the predominant species recognized for causing severe infections in neonates within the hospital environment were *Klebsiella pneumoniae* and *Escherichia coli*. The high mortality (62%) within 28 days [17] revealed the severe consequences of such infections within hospital settings.

Similarly, Labi et al. [65] reported a high prevalence of MDR-GNB in neonatal intensive care units in Ghana. Among 228 neonates, approximately 50% carried MDR-GNB, with *Klebsiella* spp. (41%) and *Escherichia coli* (26%) being the most common pathogens. The authors identified high resistance rates of these bacteria to antibiotics such as ampicillin and gentamicin. According to them, high numbers of MDR-GNB were isolated from incubators, cots, and medical equipment. Notably, MDR-GNB carriage increased with hospital stay duration, rising from 13% on admission to 91% by day 15. Odewale et al. [106] showed that high levels of MDR *Klebsiella pneumoniae* pose a significant challenge to healthcare in Southwest Nigeria. In some cases, there is a widespread occurrence of these pathogens alongside MSRA [41].

In the northern regions of Cameroon, hospital-acquired MRSA showing resistance to penicillin (91%), cotrimoxazole (87%), and tetracycline (72%) was prevalent in five major hospitals [73]. MRSA emerged from a hospital environment and has been used to indicate the persistent occurrence of nosocomial infections globally [107]. Hospital-acquired MRSA, characterized by resistance to methicillin and other beta-lactam antibiotics, is associated with severe infections such as bloodstream infections, pneumonia, surgical site infections, and soft tissue infections [73]. It is transmitted primarily through contact with infected patients, healthcare workers, or contaminated surfaces and equipment [20]. MRSA has been identified in individuals without prior contact with hospital environments [73]. The *PVL* genes mediate the virulence [20,73,74,108], and resistance is mediated by the *mecA* gene, which encodes an altered penicillin-binding protein (PBP2a) that reduces the efficacy of beta-lactam antibiotics [20,73]. Hospital-acquired MRSA also exhibits multidrug resistance, further complicating treatment [73]. Overall, the overuse and misuse of antibiotics in hospital settings facilitate the persistent spread of antibiotic-resistant bacteria and resistance genes among clinical isolates [106]. Nonetheless, several limitations hinder a thorough understanding of this critical issue in West Africa.

## 4. Challenges and Limitations Encountered by Researchers in Studying Antibiotic Resistance in West Africa

In several studies conducted in West Africa, researchers frequently conclude that antibiotic resistance is widespread and that inadequate hygiene practices may significantly exacerbate the prevalence of antibiotic resistance and its trends. However, a notable limitation of these studies is the lack of molecular techniques, which could have provided more precise insights into the genetic factors contributing to the observed resistance [46]. Indeed, apart from data paucity, the absence of molecular techniques in several studies significantly limits the in-depth knowledge of antibiotic resistance in West Africa. For instance, out of the 79 studies included in our research, only two incorporated whole-genome sequencing, and there were two studies with metagenomics. Sixty-five studies characterized antibiotic resistance phenotypically using the Kirby–Bauer disk diffusion method (n = 57) or the broth microdilution assay (n = 8), and the others were not specified.

In most cases, the bacterial isolates were characterized phenotypically using conventional biochemical techniques, which could present misleading identification as bacteria with similar phenotypes could vary genotypically [109]. Sometimes, the authors employed selective media and phenotypic confirmation to investigate the prevalence of ESBL-producing bacterial isolates. The phenotypic characterization of antibiotic resistance is crucial for understanding microbial adaptability. Yet, this approach is often time-consuming and may yield inconclusive results, as it fails to provide insights into the genetic framework underlying resistance mechanisms [110]. The genotypic identification of bacteria, employing 16S rRNA gene sequencing or other fingerprinting techniques such as matrix-assisted laser desorption ionization time-of-flight mass spectrometry (MALDI-TOF MS), presents a more robust approach for the accurate identification and characterization of bacterial isolates following cultivation [109].

Molecular methods enhance the reliability of bacterial identification by providing distinct genetic and molecular signatures that facilitate precise classification. In addition, genomic techniques can provide valuable insights into the specific genetic traits associated with the global spread of ESBL-producing bacteria and their host species [111]. Consequently, molecular techniques such as polymerase chain reaction (PCR), whole-genome sequencing and metagenomics, and DNA microarray MALDI-TOF MS are becoming increasingly used globally as they present greater accuracy and speed and are usually needed to complement phenotype characterization to develop optimal strategies for combating antibiotic resistance on time [112].

Numerous studies utilized PCR to detect specific resistance genes; however, these studies often had a limited scope and did not offer a comprehensive understanding of the antibiotic-resistance genes present in certain antibiotic-resistant bacteria. For instance, researchers usually identify extensive resistance to various antibiotics; however, their PCR methodologies frequently concentrate on specific resistance genes associated with selected antibiotics. This focus may overlook other resistance genes that could contribute to resistance against additional antibiotics, thereby limiting the understanding of the full spectrum of antibiotic-resistance mechanisms present in microbial populations. One of the studies that incorporated whole sequence analysis identified about 1131 resistance genes in 112 isolates obtained from hospitalized infants in the Gambia [17]. The other study not only detected the widespread presence of various antibiotic-resistance genes in different types of *Salmonella* spp. isolated from humans, animals, and the environment in Nigeria, but also identified genes mediated by plasmids and several virulence genes found in various Salmonella pathogenicity islands [113]. Thus, whole genome sequence analysis may help simultaneously target antibiotic-resistance genes, including plasmids, mobile genetic elements, and virulence genes. Therefore, researchers in West Africa need to integrate advanced molecular techniques into their efforts against antibiotic resistance.

While it can be argued that the challenges in tackling antibiotic resistance extend beyond genomic limitations, insufficient funding also significantly hinders research outputs in West African contexts. For example, in our review of 79 papers, 38 (48%) of the papers acknowledged their funding agency, indicating that their research was funded to some extent, three (4%) did not indicate whether they received funding, and the remaining 38 (48%) indicated that their study received no funding. Indeed, the funding situation can significantly affect research output. Many African researchers rely on limited international financing, which limits research output [114]. The studies funded in this review were also achieved through international fellowships that allow the researchers to conduct their research in internationally well-equipped research institutes. According to Caelers et al. [115], the lack of funding poses a widespread challenge for African researchers as they are often overlooked in funding initiatives. This exclusion hampers their ability to engage in innovative research that could address regional and broader global challenges [115], which, from our perspective, includes antibiotic resistance. Therefore, a multifaceted approach addressing these underlying challenges is essential for enhancing research efficacy and ultimately mitigating the impact of antibiotic resistance in West Africa and Africa in general.

## 5. Conclusions

Environmental risk factors such as environmental pollution and waste management, poor hygiene, use of antibiotics in agriculture, aquaculture, and animal farming, as well as transmission of antibiotic resistance within hospital settings, contribute significantly to the spread of antibiotic resistance in most West African countries. However, significant gaps in data, molecular characterization, and a lack of resources remain challenges to comprehensively understanding this critical public health issue in this region. An important relationship exists between environmental risk factors and the spread of antibiotic resistance. The interplay of these factors fosters the continual emergence of antibiotic resistance, which poses a persistent threat to human health. The ambient microbiome exhibits remarkable diversity, offering a vast array of genes that bacteria, particularly pathogens, can acquire across different environments. This diversity is, in fact, the most notable aspect of antibiotic resistance. Since antibiotic resistance transcends borders and can spread within and between countries, it is crucial to understand the environmental risk factors that influence its dissemination in these regions. Such understanding will aid in formulating and recommending environmental management strategies nationally and internationally to combat antibiotic resistance.

In both clinical and natural settings, the overuse or improper application of antibiotics exerts selective pressure, fostering the development of antibiotic resistance. Bacteria naturally develop resistance through genetic mutations and by acquiring resistance genes through genetic exchange [12]. Antibiotics can enhance horizontal gene transfer among pathogens, thus aiding the spread of resistance genes across different environmental sources. For example, antibiotic-resistant bacteria and their associated resistance genes found in hospital wastewater can interact with surface-water bacteria, acquiring and spreading resistance. Also, hospital and domestic wastewater, when discharged into urban water bodies, contribute to the pollution of aquatic ecosystems. Approximately 1.5 billion people in Sub-Saharan Africa depend on surface water for their livelihood [116]. This highlights the urgent need for stringent wastewater treatment and environmental monitoring to mitigate the spread of resistance from clinical to environmental settings, thereby addressing a critical dimension of the antibiotic resistance crisis.

Studies emphasize the need for enhanced monitoring in hospital environments, particularly neonatal units and hospitals, to track resistance profiles and inform treatment decisions. Monitoring antibiotic resistance in food production, hospital waste, and wastewater is also crucial. Many West African countries lack sewage treatment plants, which could significantly mitigate environmental pollution and protect public and environmental health [117]. As a result, many households depend on septic tanks with soak-away pits, often constructed just a few meters away from boreholes, wells, and other groundwater sources [118]. The increasing population in these countries has led to a significant rise in waste production, necessitating effective waste management strategies to mitigate the proliferation of diseases, particularly those associated with antibiotic resistance.

Several studies also provide a compelling case for addressing food hygiene and safety challenges in West African countries and advocate for preventive measures to curb the spread of antibiotic resistance and foodborne diseases. Surveillance should focus on the community, particularly food vendor education, addressing inappropriate, poor hygiene practices, and inappropriate antibiotic use in different environmental contexts. Continuous monitoring, infection control, and public education are key to managing antibiotic resistance in the region. Lastly, more genomic approaches and resources are needed to track resistant strains and resistance genes while integrating national antibiotic plans with global systems.

## Figures and Tables

**Figure 1 microorganisms-13-00951-f001:**
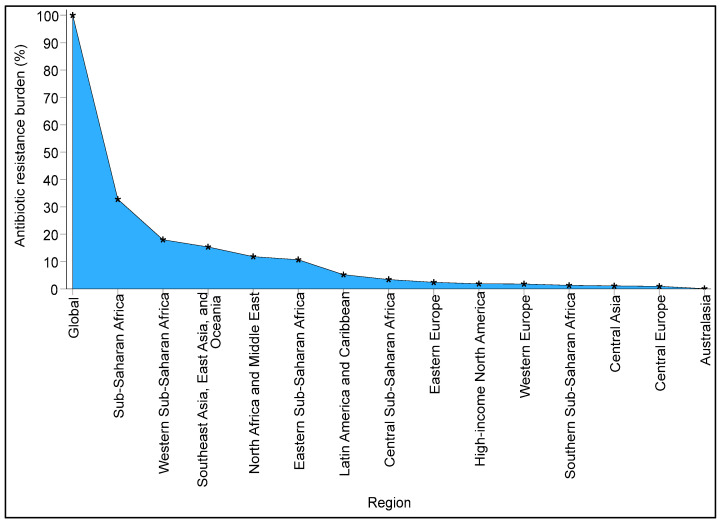
The global burden of antibiotic resistance is greater in Sub-Saharan Africa, particularly in the Western Sub-Saharan region, than in other regions. The raw data were obtained from the repository of Murray et al. [4], and the percentages for each region were derived from the global estimate.

**Figure 2 microorganisms-13-00951-f002:**
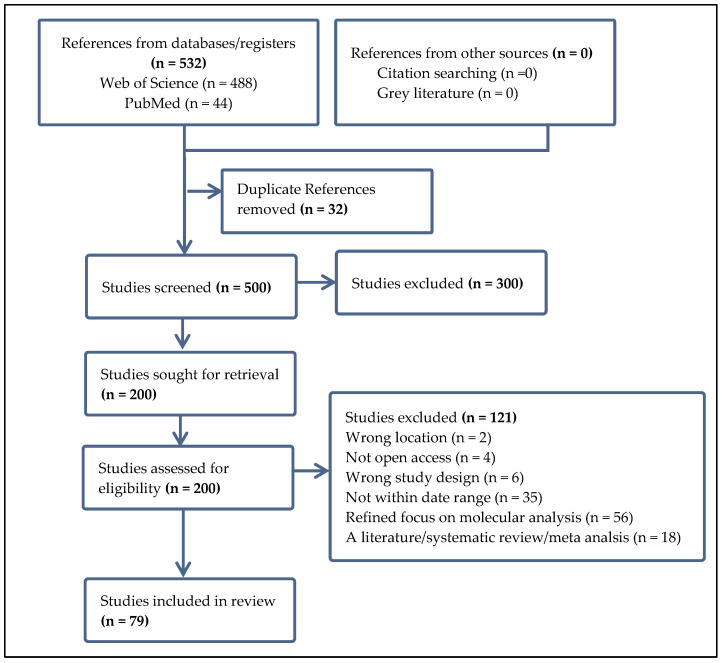
The workflow of the articles included in this review. Studies were exported from the Web of Science and PubMed databases and screened for eligibility based on our inclusion and exclusion criteria.

**Figure 3 microorganisms-13-00951-f003:**
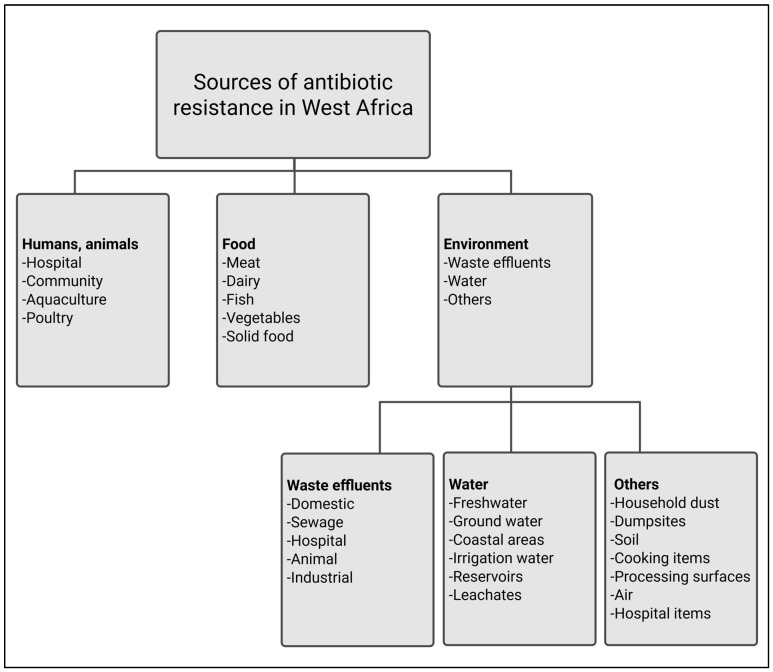
Observed sources of antibiotic resistance in West Africa.

**Figure 4 microorganisms-13-00951-f004:**
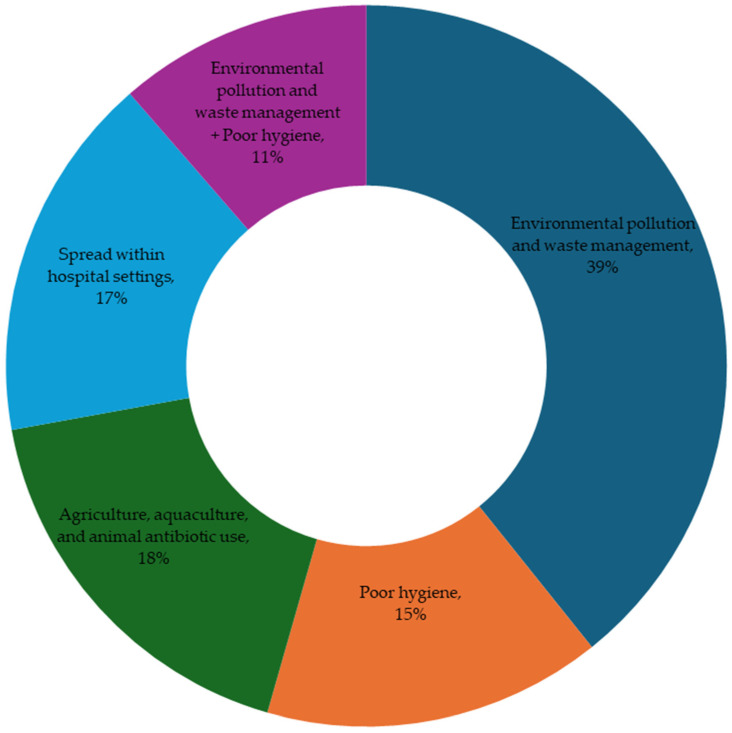
Environmental risk factors contributing to the spread of antibiotic resistance in West Africa. The percentages are the frequencies at which we see each factor in the papers reviewed.

**Figure 5 microorganisms-13-00951-f005:**
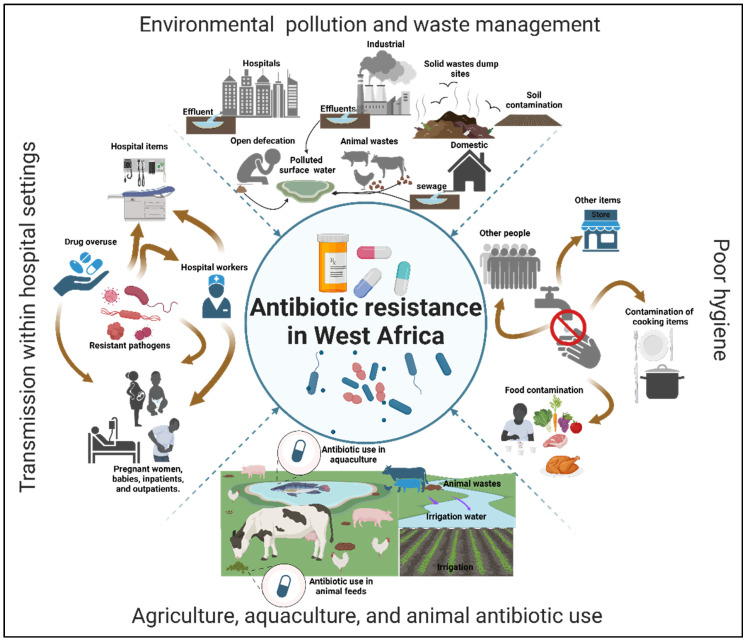
Most studied environmental risk factors facilitating the spread of antibiotic resistance in West Africa.

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
