# Peer review of "Environmental Risk Factors Contributing to the Spread of Antibiotic Resistance in West Africa"

_microorganisms, 2025, doi:10.3390/microorganisms13040951_

Round 1
Reviewer 1 Report
Comments and Suggestions for Authors
I would like to express my gratitude to the Editor for the opportunity to review this manuscript. The article presents a rich and highly relevant compilation of information directly aligned with the topic of antibiotic resistance in the West African context.
The language used throughout the manuscript is generally of good quality; however, certain sections could benefit from minor linguistic improvements to enhance clarity and flow. I was particularly pleased to observe that the literature cited in the manuscript is relatively recent and includes references to studies employing molecular microbiology methods, which adds significant depth and scientific rigor to the work.
While reviewing, I identified a few minor errors: specifically, the omission of a period after the reference in line 54, and the lack of italicization for bacterial species names in line 289. Despite these minor issues, the manuscript as a whole represents an important and timely review of the growing challenge of antibiotic resistance in West Africa.
With minor revisions, I recommend this article for publication.
Author Response
Comment 1: While reviewing, I identified a few minor errors: specifically, the omission of a period after the reference in line 54, and the lack of italicization for bacterial species names in line 289. Despite these minor issues, the manuscript as a whole represents an important and timely review of the growing challenge of antibiotic resistance in West Africa.
Response 1: Thank you for your comment. We have corrected the errors in line 54 and italicized the bacterial species names in line 289.
Reviewer 2 Report
Comments and Suggestions for Authors
This manuscript gives an overview about antibiotic resistance in West Africa. Topic of this manuscript is an interesting issue, however, the text is very general. Some specific data should be added to the text.
1) I suggest to authors to add some prevalence data of ESBL producing K. pneumoniae, and other ESBL producing Enterobacterales from West Africa.
2) Some prevalence data about carbapenem resistant Enterobacterales, Pseudomonas aeruginosa, Acinetobacter baumannii from West Africa should be added to the manuscript.
3) MRSA is mentioned in the text however, some prevalence data from West Africa should be added to the text.
4) Prevalence data from West Africa about clinical isolates, or veterinary isolates or enviromental isolates would be useful to be added to the text.
5) The prevalence data should be demonstrated on a diagram or in a table.
Author Response
Comments 1: I suggest to authors to add some prevalence data of ESBL producing K. pneumoniae, and other ESBL producing Enterobacterales from West Africa.
Response 1: Thank you for your insightful comment. According to the papers we reviewed, we have incorporated data regarding the prevalence of ESBL-producing Klebsiella pneumonia and other ESBL-producing Enterobacterales into our supplementary table. Additionally, we have included some prevalence data within the main text. Please refer to lines 277 to 291 of the updated manuscript.
Comments 2: Some prevalence data about carbapenem resistant Enterobacterales, Pseudomonas aeruginosa, Acinetobacter baumannii from West Africa should be added to the manuscript.
Response 2: Please see supplementary Table 1
Comments 3: MRSA is mentioned in the text however, some prevalence data from West Africa should be added to the text.
Response 3: Based on the findings from the papers we reviewed, we addressed the prevalence of MRSA in our manuscript. and included this information in our supplementary table.
Comment 4: Prevalence data from West Africa about clinical isolates, or veterinary isolates or enviromental isolates would be useful to be added to the text
Response 4: We incorporated the necessary bacterial isolates in each section (see sections 3.3.1 to 3.3.4). Discussing the prevalence of clinical, veterinary, or environmental isolates at length would dilute the message we aimed to convey. However, we have included the prevalence of all antibiotic-resistant bacteria and their types in the supplementary table.
Comments 5: The prevalence data should be demonstrated on a diagram or in a table.
Response 5: Please see supplementary Table 1